# Analysis of Relationships and Sustainability Performance in Organic Agriculture in the United Arab Emirates and Sicily (Italy)

**Khalid Butti Al Shamsi** [1], **Paolo Guarnaccia** [1], **Salvatore Luciano Cosentino** [1],
**Cherubino Leonardi** [1], **Paolo Caruso** [1], **Giuseppe Stella** [2] **and Giuseppe Timpanaro** [1,*]

[1]  Dipartimento di Agricoltura, Alimentazione e Ambiente (Di3A), University of Catania, Via S.Sofia, 100, 95123 Catania, Italy; kbalshamsi@gmail.com (K.B.A.S.); paolo.guarnaccia@unict.it (P.G.); cosentin@unict.it (S.L.C.); cherubino.leonardi@unict.it (C.L.); pcamps@libero.it (P.C.)

[2]  PH3DRA Laboratories, University of Catania & INFN, Via S.Sofia, 64, 95123 Catania, Italy; giustella75@gmail.com

*  Correspondence: giuseppe.timpanaro@unict.it; Tel.: +39-095-758-0305

**Abstract:** Climate change, recurrent economic and financial crises and food security issues are forcing society to look at the increasingly widespread use of "sustainable" production practices. These are often translated into innovations for businesses that are not always easily achievable other than through specific investments. This work sets out to assess the sustainability performance of organic farms, which represent a sustainable production model in terms of values, standards, practices and knowledge on the ground. The research was carried out in two geographical contexts (the United Arab Emirates (UAE) and Sicily, Italy) which have certain environmental and socio-economic issues in common, particularly in productive sectors representative of organic agriculture. This was done with the help of the Food and Agriculture Organization of the United Nations (FAO) Sustainability Assessment of Food and Agriculture Systems (SAFA, in the rest of the text) framework and social network analysis to study the sustainability performance of organic farms within non-structured local production systems in the form of formal enterprise networks that, on the contrary, operate with a recognized and common aim. The results demonstrate both their attainment of excellence and the existence of criticalities, thus, identifying routes to possible improvement.

**Keywords:** SAFA–FAO; social network analysis; informal relationships; organic agriculture; food security; sustainability

## 1. Introduction

In the literature, many contributions have focused on an assessment of the sustainability performance of farms using a wide range of measurement tools [1], the issues involved in assessing the environmental impacts of agriculture [2], and on the implications that sustainability has for the strategic decisions of agricultural businesses [3]. The tools proposed for these assessments are usually based on indicators that are incapable of analyzing the complexity of agricultural practice; they also differ in their scoring and aggregation methods, in their time frames of observation, in their input of the data observed, and in the complexity of using and understanding them, and above all, in reconciling the value judgements of those developing the tools and those of the farmers themselves. Hence, a number of studies have focused on criticism of these tools, and highlight that the direct measurement of productive practices does not always make it possible to assess the environmental impact of agricultural activities (for example, in terms of nitrogen loss and the risk from pesticides). They also suggest that it is more appropriate to integrate direct and indirect assessments by using

models based on macro data (regarding the soil, climate, growing systems, etc.) available at area level. For these reasons, best practices such as prior planning of each indicator's practical objectives, of the end users, of the spatial and time scales to be used, and of the correct definition of the reference values are often recommended.

Finally, complaints have been made regarding the low contribution of research to the links between the strategic decision-making process and the assessment of sustainability in farms. In operational reality, it turns out that the choice of more sustainable agricultural practices often requires adequate prior assessments to identify, measure, evaluate and communicate sustainable development. The use of tools for assessing sustainability within the decision-making and management processes of agricultural businesses is often constrained by the complexity of the systems proposed, hesitation in applying the results of the measurement, a lack of flexibility and options offered, and by limited feedback and lack of communication from the market in which the company operates, which eventually nullifies the movement towards sustainable agriculture.

In late 2015, a collaboration was set up between the Food and Agriculture Organization of the United Nations (FAO), the International Federation of Organic Agriculture Movements (IFOAM) and the PhD course in Agricultural, Food, and Environmental Science of the University of Catania with the objective of measuring agricultural sustainability, food security and sovereignty in different geographical contexts and assessing the outlook for adopting "organic" production and consumption models. The FAO, in particular, has for some time produced and distributed Sustainability Assessment of Food and Agriculture Systems (SAFA), a specific open source software (Version 2.2.40, on: www. fao.org/nr/sustainability/sustainability-assessments-safa, FAO Rome, Italy) package for measuring sustainability in agri-food production chains used in this study.

The organic production method was chosen because, as is well-known, it is a production system that tends to encourage the building of relationships between individuals, enterprises and institutions (public and private) positioned upstream, level with, and downstream of production. These relationships may or may not be strictly codified within the supply chain and address the productive, strategic and cognitive spheres in order to confront and/or resolve specific production and market issues (problem-solving with the exchange of raw materials, the sub-contracting of processes, the exchange of semi-finished products and services, etc.).

A number of studies have demonstrated that the existence of a network of relationships between actors connected in different ways creates value for them because it enables them to combine the different knowledge assets of the different partners [4], and increases the value of investment in research and development. This process appears to be particularly important for small enterprises since it is unlikely that they possess the resources to master the skills needed to compete inn markets [5]. Enterprises, however, do not necessarily have to be bound by formal links in order to enjoy the transfer of knowledge since another triggering factor is "proximity" between actors in a geographical area. This favors mechanisms of cultural socialization, the sharing of values, standards and language, i.e., organizational culture pertaining to the cognitive dimension of social capital [6,7].

The nature and quality of direct and indirect relationships in organic agriculture is thus able to produce a multiplier effect on the sustainability performance of enterprises and on their ability to contribute to sustaining local production systems, which affects food security and sovereignty. The attainment of a higher level of sustainability in the production process represents a real innovation for the enterprise—this often translates into a set of organizational and management changes that are essential for the implementation, attainment and maintenance of a specific sustainability performance.

Two areas were chosen, the UAE and Sicily, the former because it has launched a food policy to encourage land use and the adoption of sustainable practices like organic farming in order to counter a scarcity of natural resources, excessive immigration from different geographical areas and a balance of trade in disequilibrium; the latter because despite having the typical food security issues of areas with high urbanization, it has always been inclined to the use of organic farming (in terms of the area dedicated to this it ranks first in Italy and second in the number of operators). Therefore, Sicily

is a useful benchmark for the UAE. The two areas share a number of pedo-climatic (climate, rain, temperature, soil characteristics, etc.), production, demographic and migratory characteristics, as well as similar food-related issues (high levels of food insecurity, household spending capacity, obesity rates, etc.) and a clear desire to pursue sustainable development objectives [8].

For these reasons, a group of organic farms representative of the main production sectors (fruit and vegetables, arable, tropical and subtropical fruit farming, etc.) in the two geographical contexts was selected and studied using the FAO's SAFA [9]. These are organic enterprises belonging to networks that share a common desire to improve their sustainability performance to the extent of joining the SAFA monitoring process. Their market positioning is through hybrid channels with a tendency to favor "farmer's markets", and they appear motivated to use the strategic lever of sustainability to communicate their performance to buyers.

The objective of this study, using social network analysis (SNA) methods, is to analyze the network of farms taking part in the SAFA program, which are linked by shared company strategies regarding good governance, the environment, economic resilience and social welfare. The aim is to identify common (or less common) weaknesses or strengths and to propose improvements to their approach to sustainability via the organic production method.

Ultimately, therefore, once the hypothesis had been established that the network (formal or informal) of organic farms influences sustainability, measurements were taken of the state of the network (i.e., the geographical differences between farms, the sharing of information, etc.) on the one hand, and of sustainability performance using SAFA on the other, to then highlight whether the state of the network is linked to sustainability performance and, if so, in which areas.

## 2. Conceptual Framework

Sustainability assessment has been a field of study and research for several years with the aim of supporting decision-making and environmental, economic and social policy. The literature is rich with models and evaluation indicators that can be differentiated by their ontological, methodological and epistemological point of view. The application of methods and tools is also very diverse in real decision-making contexts [10].

Since 2009, the FAO has been preparing guidelines (SAFA) to assess the overall impact that food and agriculture-related initiatives along the entire food supply chain have on the environment and on people [11]. The objective of the FAO was to create a comprehensive framework to unite the different experiences and methodological approaches of the various disciplines (biology, economics, ethics, the environment, etc.) and actors (the political and productive world, civil society, scholars and researchers) to sustainability. A participatory development model has led to the creation of SAFA, constituting a first step towards the international harmonization of requisites favoring the sustainable production and retail sale of food and agricultural products.

SAFA has already had food security applications at the production sector level in an evaluation carried out on the sustainability of small-scale livestock farms in Indonesia, a country in which local policy is aimed at promoting sustainable agricultural practices in small family farms [12]. Other applications include coffee production in Uganda, with measurements to assess the effects of organic certification, which resulted in positive conclusions on the effect on cooperation, governance and the main social, environmental and economic aspects of sustainability [13]. Related to this, an analysis of banana production in Costa Rica was carried out to holistically assess the effective sustainability performance of this production system, and to identify and launch optimization initiatives for more sustainable production [14]. The FAO framework was also utilized in a multi-dimensional, systematically assessed understanding of the concept of "sustainable development" in poultry production, with findings of particular interest given the importance of this sector to the worldwide food system [15]. Numerous potential pathways to sustainable development in poultry production were identified, linking elements such as the welfare of animals and workers (social aspects),

biodiversity (environmental aspects), governance of the food chain (institutional aspects) and the development of poultry as a high-value food throughout the world (economic aspects).

Analyses of production systems using SAFA have been carried out with regard to Hungarian organic farming [16], aimed at defining a conceptual framework for measuring all aspects of sustainability among a sample of farms. Also, an evaluation of agri-environmental indicators to guide the decisions of agricultural operators through agricultural policy initiatives was also recently carried out in the Czech Republic [17], where methodological difficulties in introducing and implementing sustainable farming practices emerged. An attempt to achieve a common understanding of how to measure sustainability in the food sector was carried out in Europe and Mexico among a sample of 60 agri-food enterprises, demonstrating the practicability of the SAFA tool even in difficult circumstances (58 sustainability targets classified into 21 themes and four dimensions), involving each company choosing an individual set of appropriate indicators and a variable evaluation questionnaire length according to the size and complexity of the enterprise [18].

A comparison between different approaches to the assessment of sustainability in farms, agricultural systems and supply chains has led to the identification of significant differences between SAFA and other models in terms of the scope, level of assessment and precision of the indicators used. Moreover, occasional contradictory results have suggested the advisability of including a precise definition of the notion of "sustainability" together with a description of the methodology and indicators in order to achieve the harmonization of indicators and hypotheses [19]. For this reason, SAFA turns out to be a useful benchmark paradigm for highlighting the differences between the different approaches and for making the assessment results more comparable [20]. A comparison of several applications for the assessment of sustainability (IDEA—Indicateurs de Durabilité des Exploitations Agricoles or Farm Sustainability Indicators; RISE—Response-Inducing Sustainability Evaluation model; SAFA; SOSTAR - analysis of farm technical efficiency and impacts on environmental and economic sustainability, MOTIFS—Monitoring Tool for Integrated Farm Sustainability and 4Agro) at farms in Northern Italy has made it possible to study the choice of indicators, the availability of data and the involvement of stakeholders, as well as to add more evaluation scales for agriculture (the environment, society, the economy and governance) [21–23].

Another interesting work aimed at f defining the contribution of organic food systems to sustainability used the SAFA guidelines applied at operator, product and spatial/political level, as well as three sustainability strategies relating to efficiency, consistency and sufficiency as a framework of reference [24,25]. It showed that organic food systems can provide sufficient food if demand patterns shift towards products that consume less resources (that is, via dietary patterns and food waste). This confirms the importance of the social dimension in the biological system, and that innovation and further development of the organic system are essential in tackling future challenges.

Social network analysis views social relationships in terms of network theory and consists of nodes and ties (also called edges, links, or connections). Nodes are the individual actors within the networks, and ties are the relationships between the actors. It measures networks of people and helps evaluators determine how people are connecting and around what issues and projects.

Collaboration processes and social dimensions are essentially relational in nature: they require the creation and maintenance of a connection between one or more actors or organisations. Given the relational nature of network activities, social network analysis (SNA) offers a framework to study and model different aspects of agricultural innovation and scaling [26–28]. SNA enables a better understanding of the complexity and multi-dimensionality of multi-stakeholder innovation processes [29,30].

## 3. Materials and Methods

### 3.1. Research Design

Organic agriculture is a sustainable agricultural model by definition, the aim of which is to create integrated, human, ecological and economically sustainable agricultural systems based in particular, on renewable local resources and on the management of ecological and organic processes. It is also sustained by relationships between producers and between them and the local area and institutions (technical assistance services, local market representatives, consumer associations, research centers, etc.), and as such, can activate a beneficial process that leads to an increase in the sustainability performance of the entire production system.

Despite being so important, few studies have focused on the measurement of relationship systems between organic farming enterprises and how these can affect their sustainability performance, other than within the limits of the stipulations of community regulations governing organic production. With this aim, two geographical areas were selected with similar environmental and socio-economic issues, namely, the UAE and Sicily, where organic agriculture is particularly widespread and of high economic importance.

A sample of organic enterprises was selected within these areas, and this sample was assessed in terms of sustainability using the FAO SAFA application.

A sample of 16 enterprises was selected in the two production systems, taking account of:

- production sector representative of local organic agriculture;
- size of the farm/enterprise;
- the willingness of the enterprise owner to join a network specifically aimed at measuring sustainability using SAFA, providing data on the organization and management of the enterprise also as regards accounting aspects;
- active role in the local market;
- clear demonstration of economic and social resilience.

The farms were identified using a stratified random number system. The chosen farms were integrated into the network by organizing two workshops. During the first workshop (at the end of 2015)—attended by 30 farms—the aims of the study were presented, the different aspects of sustainability to be measured were analyzed, and the nature and meaning of the indicators was explained as well as the possibility of committing to a common network set up by SAFA and to collaborate by exchanging information or commercial or technological activities, i.e., by performing one or more common activities in the relevant organic production chain. The second workshop in 2016 was only attended by the 15 businesses that had committed themselves to measuring sustainability with SAFA, to choosing a process for optimizing resources and relational skills and to sharing common activities by subdividing the costs (e.g., trade fairs, commercial networks with local markets, ethical purchasing groups, etc.), while maintaining their freedom and autonomy as individual businesses.

The fieldwork was carried out from September to November 2017 and was repeated over a period of 2 months to pick up any changes in the sustainable management of activities. The SAFA survey was conducted at the individual farms that accepted the experimentation. To this end, the questionnaire was administered and detected all variables listed in the Appendix A. To reduce the arbitrariness of the survey, 2 interview teams were formed, one for Sicily and the other for the UAE. The questionnaire had been previously tested and engaged the participants for on average 8 h, to allow the entrepreneurs time to search for documentation to support their answers.

### 3.2. Measurement Tools: The SAFA Approach

According to SAFA, sustainability involves 4 interrelated dimensions, namely "Good Governance" (also "corporate" ethics), "Environmental Integrity", "Economic Resilience" and "Social Well-Being". As well as the strictly environmental dimension, the modern concept highlights that sustainability is

achieved through the virtuous and balanced management of human and ecological resources in its broad sense, using an approach defined as "holistic". SAFA includes as many as 116 detailed indicators across 21 themes (or sustainability "goals"), and 58 sub-themes (more detailed objectives). Moreover, it is also possible to choose the appropriate indicators for the specific context as well as sustainability performance ratings.

SAFA is aimed at producers and other operators within the agri-food system, who can in this way assess their performance and/or plan the targeted use of natural resources and/or develop partnerships with suppliers in order to improve their socio-economic and environmental performance (business-to-business and business-to-consumer strategies). It is also aimed at consumers, who can form a critical approach to consumption, and policy-makers engaged in defining sustainable development strategies, in evaluating the externalities of production processes and in the governance of shared policies between the actors involved (institutions, economic actors, civil society and citizens).

The SAFA questionnaire addresses a number of aspects and has been subdivided into macro areas or contexts, themes and sub-themes (with each assigned an ID), as shown in the Appendix A.

The results of the 15 interviews were entered into the SAFA tool software (Version 2.2.40) developed by the FAO [31]. The data entered into the system by each farm and relating to the individual sub-themes was translated into ratings. For the purposes of the SNA analysis, a value of between 0 and 5 was associated with each rating. Additionally, each rating was weighted according to the level of accuracy of the data given (Table 1).

**Table 1.** Accuracy score and data quality per indicator in SAFA (Version 2.2.40).

| Rating | Value | Accuracy Score | Value |
|--------|-------|----------------|-------|
| Best | 5 | Low quality data | 1 |
| Good | 4 | Moderate quality data | 2 |
| Moderate | 3 | High quality data | 3 |
| Limited | 2 | - | - |
| Unacceptable | 1 | - | - |
| Not relevant | 0 | - | - |

### 3.3. Analysis Methodology

To analyze social structures and to measure actor attributes social network analysis [32] was used; it employs matrix calculations and the representation of relationship characteristics through graph theory [33]. In particular, a graph is defined as a set of ordered pairs

$$G = (V, A) \tag{1}$$

consisting of vertices n (nodes) and arcs m (or bridges) that connect them.

SNA was chosen for its ability to explain the level of relevance of actors within a given network structure or their degree of centrality in the network [34].

The first level of analysis, represented by the "Structure" factor, was assigned to analyze the effects of the overall social network structure, not only on the ability of individual actors to initiate processes to achieve a higher level of sustainability but also on their ability to assume pre-eminent positions and prestige roles within the network. The second level, on the other hand, studied the relevance (centrality) of the actor and its role in the network (role).

The intention was thus to take into account the ability of the overall network to facilitate access to all of the knowledge assets regarding sustainability contained within it. Another relevant indicator is the density of the network [33], which measures the number of relationships within the network as a proportion of those that, given the number of nodes, could potentially exist. In particular, the degree of density (which assumes values of between 0 and 1) can be defined as:

$$D = \frac{2a}{n(n-1)} \tag{2}$$

in which $a$ is the number of active relationships and $n$ is the number of nodes in the network. The degree of centrality expresses the number of relationships (or their relative importance, if expressed as intensities) referring to a given node [35]. For the i-th node in particular, the degree of centrality can be defined as that:

$$DC_i = \sum_{k=1}^{N} h(n_i, n_k)(N-1)^{-1} \tag{3}$$

in which $h$ has a non-null value if the arc connecting the i th node with the k th node is active.

The centrality of the actor within the network was measured using differing indicators depending on how the concept of pre-eminence was intended to be expressed, although the most widespread of these is "degree centrality" [35]. This is based on the actor's level of activity in the network, measured on the basis of the number of relationships it establishes, or also, of its degree of popularity among the other actors.

$$AC_i = \frac{1}{\lambda} \sum_k a_{k,i} X_k \tag{4}$$

where $a_{k,i}$ is the adjacency matrix of the network, $x_k$ denote the score of the $k$ th node and $\lambda$ is a constant. Hence, if the $a_{k,i} = 1$ if $k$ th node is adjacent to i th node, and $a_{k,i} = 0$ otherwise.

Instead the "closeness centrality" indicator [36] is based on the concept that the pre-eminent actors in the network are those that can most easily transfer information to all the others; these actors also have the advantage of being able to learn the new cognitive resources developed in the network more rapidly and more easily [37]. In network theory this is defined as the mean geodesic distance (i.e., the shortest path) between a vertex $v$ and all other reachable vertices. It is a measure of how near or far pairs of consumers are in their behavior and/or choices. "Eigenvector centrality" is a measure of the importance of a node in a network. It assigns relative scores to all nodes in the network based on the principle that connections to high-scoring nodes contribute more to the score of the node in question than equal connections to low-scoring nodes [38]. It is a measure of how much a consumer can influence the choices and behaviors of others within a network. Finally, the "betweenness centrality" indicator measures the ability of a single actor to directly influence the transit of information within the network, thus influencing the behavior of the other actors in the network and the development of the network itself [35]. This last indicator of centrality also takes the overall structure of the network into account because it assigns more central positions to those actors connecting parts of the network that are otherwise detached [39]. It is considered as a measure for quantifying the control of a human on the communication between other humans in a social network.

The elaborations were performed with UCINET software (version 6.4, Analytic Technologies, Lexington, KY, USA) for Windows, developed by Freeman et al. [40]. It can handle over 30,000 nodes, release graphics and perform solid matrix analysis such as matrix algebra and multivariate statistics.

## 4. Analysis of the Results of the FAO SAFA Application in the UAE and Sicily

### 4.1. Characteristics of the Farms

Sicily and the UAE are two geographical areas that share a number of common issues:

- similar climates that in some contexts influence the type of agriculture practiced;
- limited availability of the quality and quantity of water resources needed for use in some production categories;
- widespread sensitivity to the use of organic farming methods;
- a social population structure that due to certain international economic dynamics and policies, (economic crises and a lower availability of income for the qualitative/quantitative satisfaction of

basic needs such as food' migratory flows that affect certain aspects of diet and of the organization of the area's social, cultural and working system, etc.) runs the risk of being severely affected by problems of under-nutrition.

According to the most recent official statistics (Table 2), organic farming in the UAE involves 1.2% of the cultivated land area (about 4.6 hectares and 92 farms, FiBL 2018), with marked rates of growth in the past 10 years (+4.585). Sicily, on the other hand, has a longer-standing tradition in organic agriculture (+244 in the past decade), involving 31% of the agricultural area being used (over 427 thousand hectares and 9.4 thousand farms, SINAB 2018).

**Table 2.** Organic farming in the United Arab Emirates and Sicily (*).

| Indication | UAE | | | | Sicily | | | |
|---|---|---|---|---|---|---|---|---|
| | Source | Date | Unit | Value | Source | Date | Unit | Value |
| Organic Agricultural Land | FiBL, 2018 | 2016 | hectares | 4590 | SINAB, 2018 | 2017 | hectares | 427,294 |
| Organic shares of total agricultural land | FiBL, 2018 | 2016 | percent | 1.2 | SINAB, 2018 | 2017 | percent | 31.1 |
| I year growth | FiBL, 2018 | 2016 | - | +304 | SINAB, 2018 | 2017 | | +17.5 |
| 10 years growth | FiBL, 2018 | 2016 | - | +4585 | SINAB, 2018 | 2017 | | +244 |
| Operators | | | | | | | | |
| -producers | FiBL, 2018 | 2016 | n. | 92 | SINAB, 2018 | 2017 | n. | 9385 |
| -processor | FiBL, 2018 | 2016 | n. | 6 | SINAB, 2018 | 2017 | n. | 2223 |
| -importers | FiBL, 2018 | 2016 | n. | - | SINAB, 2018 | 2017 | n. | 18 |
| -exporters | FiBL, 2018 | 2016 | n. | 7 | SINAB, 2018 | 2017 | n. | - |

(*) Our elaboration.

The samp is equally distributed between the two areas, as can be seen in Table 3. It is representative of the structural characteristics of organic farming in Sicily and the UAE. The farm areas vary between a minimum of 2 hectares and a maximum of 70 hectares in Sicily and between 2 hectares and 5 hectares in the UAE, with averages of 34 and 3.5 hectares, respectively. The predominant production sectors are vegetables (75% of the sample) and livestock (44%), but also citrus fruits and cereal (each 25%) and to a lesser degree fresh and dried fruit, wine growing, etc.

In terms of human capital, outsourced staff predominate, both permanent and occasional. For these two types of work an average of 12 and 10 workers are employed, respectively, with up to a maximum of 55 employees. The highest amount of activity was found in intensive production sectors (horticulture in greenhouses).

The farms in the sample are mainly active in their local market (an average of 62% in Sicily and 73% in the UAE), although international export destinations are also significant (13% of the entire sample). However, the existence of hybrid methods of selling products to the market should be noted, with a simultaneous opting for futures markets as well as other destinations [41].

Finally, as regards activities undertaken to increase the resilience of the production system, by converting to organic methods farms have achieved an overall increase in the sustainability of their production, integrating activities to improve water and soil management with steps to protect biodiversity and the landscape (56% have introduced crop diversification).

"Warning system" attitudes are also widespread. shown by a quest for innovation in the field of mechanization and also through the introduction of forms of sharing to facilitate the use of modern technologies at low cost. as well as other initiatives to improve farm management (56%). Optimized management of farm systems is also common; this is fundamental in maintaining the survival and competitiveness of the business. with investments made so that changes can be managed at different levels in a non-traumatic way (about 63% showed interest both in sharing good practices and in connecting with the world of research). Finally. the diversification of production activities by creating supply chains (e.g., production of forage and related livestock breeding) was triggered by market analysis activities (82% of cases).

**Table 3.** Main characteristics of organic farms detected in UAE and in Sicily Italy (2017) (*).

| Farms, n. | Localization | Total Surface, ha | Types of Production | | | | | | | |
|---|---|---|---|---|---|---|---|---|---|---|
| | | | Citrus Fruit | Vegetable | Cereals | Fresh Fruit | Died Fruit | Grapes and Wine | Grazing | Livestok Farm |
| | | | % Surface or n. Units of Animal Heads | | | | | | | |
| 1 | Carlentini, Italy | 36.0 | 40.0 | 60.0 | - | - | - | - | | - |
| 2 | San Cataldo, Italy | 32.0 | - | - | 100.0 | - | - | - | | 40, meat cows + 50, pigs |
| 3 | Butera, Italy | 11.0 | - | - | 10.0 | 10.0 | 5.0 | 75.0 | | - |
| 4 | Aidone, Italy | 21.0 | - | - | 40.0 | - | - | - | | - |
| 5 | Acate, Italy | 70.0 | 20.0 | 80.0 | - | - | - | - | | - |
| 6 | Catania, Italy | 32.0 | 40.0 | 60.0 | - | - | - | - | | - |
| 7 | Belpasso. Italy | 70.0 | 25.0 | - | 65.0 | - | 10.0 | - | 10.0 | 80, sheep |
| 8 | Catania. Italy | 2.0 | - | 60.0 | - | - | 40.0 | - | | - |
| 9 | Dubai. UAE | 2.7 | - | 100.0 | - | - | - | - | | - |
| 10 | El-hain. UAE | 5.0 | - | 100.0 | - | - | - | - | | 30, sheep; 20, goats |
| 11 | El-hain. UAE | 5.0 | - | 100.0 | - | - | - | - | | 50, meat cows |
| 12 | Abu Dhabi. UAE | 2.0 | - | 100.0 | - | - | - | - | | 20, meat cows; 60 pigs |
| 13 | Al ghaidi. UAE | 4.5 | - | 100.0 | - | - | - | - | | - |
| 14 | Abu Dhabi. UAE | 5.0 | - | 100.0 | - | - | - | - | | 100, sheep |
| 15 | Abu Dhabi. UAE | 2.0 | - | 35.0 | - | 65.0 | - | - | | - |
| 16 | Abu Dhabi. UAE | 2.0 | - | 100.0 | - | - | - | - | | 50 sheep; 50 goats |

| Farms. n. | Permanent Workers. n. | | Temporary Workers. n. | | Reference Market % | | | Resilience Actions |
|---|---|---|---|---|---|---|---|---|
| | Family Workers | Non-Family Workers | Family Wrkers | Non-Family Workers | Local | National | International | |
| 1 | 3 | 12 | - | 13 | 65 | 4 | 31 | A + D + E + F |
| 2 | 2 | 5 | - | 0 | 45 | 30 | 25 | A + B + D |
| 3 | 2 | 7 | - | 15 | 75 | 0 | 25 | A + C + D + E + F |
| 4 | 1 | 1 | - | 5 | 50 | 30 | 20 | D + E + F |
| 5 | 2 | 19 | - | 11 | 70 | 0 | 30 | A + C +E + F |
| 6 | 1 | 2 | - | 4 | 75 | 20 | 5 | B + D + F |
| 7 | 1 | 1 | - | 5 | 35 | 35 | 30 | A + F |
| 8 | - | 6 | - | - | 85 | 15 | 0 | B + C + D + E + F |
| 9 | - | 25 | - | - | 80 | 10 | 10 | A + C |
| 10 | - | 20 | - | 20 | 60 | 35 | 5 | C + D + F |
| 11 | - | 20 | - | 30 | 55 | 45 | 0 | A + B +E + F |
| 12 | - | 9 | - | - | 75 | 25 | 0 | C + D + E + F |
| 13 | - | 55 | - | - | 70 | 10 | 20 | A + C + E |
| 14 | 1 | 6 | 1 | - | 90 | 10 | 0 | B + D + F |
| 15 | - | 2 | - | - | 95 | 5 | 0 | A + C + E + F |
| 16 | - | 5 | - | - | 65 | 25 | 10 | B + C + D + E + F |

(*) Our elaboration. Resilience actions are as indicated = A: Crop diversification; B: Diversification of productive activities; C: Adoption of warning systems; D: Exchange of information on good practices; E: Connection with the research world; F: Market analysis.

### 4.2. Analysis of the Affiliation Network

During the first stage, weighted affiliation matrix was built in which the farms participating in the SAFA program were shown on each row, ith the columns representing the sub-themes. Each element in the matrix showed the value attributed to the information given by the farms specifically in relation to a given sub-theme. Thus, his matrix was able to link the actor (farm) in the network to an event (quantified information).

The affiliation network graph created in this way (Figure 1) shows yellow graphic elements to indicate the farms (actors) and colored graphic elements to indicate the events, n other words the responses related to the macro-theme. In particular, he "Good Governance" macro theme is shown in blue, hat of "Environmental Integrity" in green. "Economic Resilience" in red, nd finally, he "Social Well-Being" theme in orange.

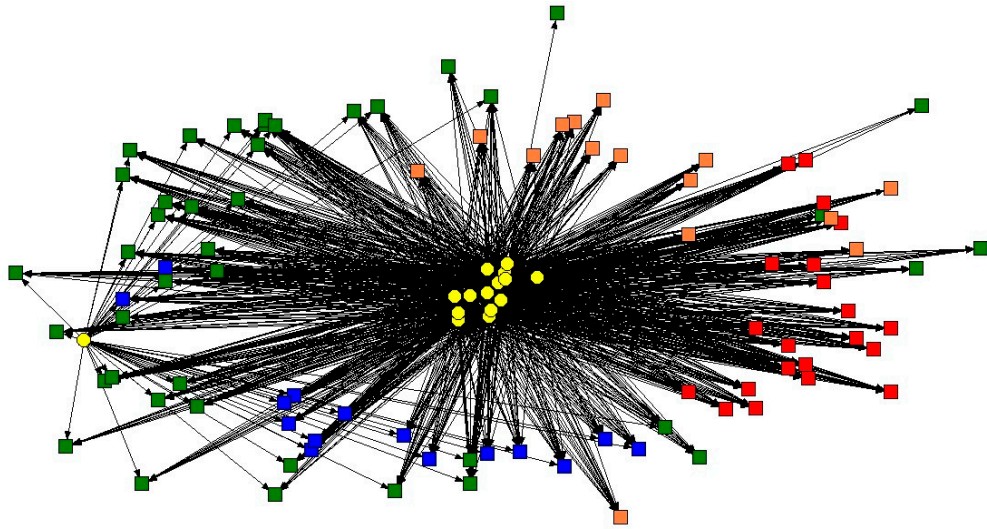

**Figure 1.** Graph of the affiliation network in the sample of organic farms that joined the measurement of sustainability with SAFA–FAO in the UAE and in Sicily. Italy (2017). The yellow graphic elements indicate the farms (actors) and the colored graphic elements indicate the events (responses related to the sustainability macro-theme).

The link between actors and events represents the sharing of a certain opinion rating in relation to a sub-theme addressed, while lengths and thicknesses are correlated to the rating score. In precise terms, the length is inversely proportional to the rating score while the thickness is directly proportional.

From an overall glance at the affiliate network it can be deduced that the aspects relating to the environment have lower rating scores on average than the others; they are in fact in a marginal area of the graph whereas aspects relating to good governance and social issues have higher ratings.

In the sphere of "environmental integrity" the farms showed sensitivity to the theme of air pollution, but the low values of the related indicator can be traced to a lack of tools able to measure the actual pollution generated by their production activities. As for livestock farms, these demonstrated better results in relation to the quality and use of water—all the operators stated that their waters were not contaminated either by livestock raising or arable activities. Only livestock farms were asked questions related to animal welfare and all achieved excellent results, stating that they prefer preventative actions for livestock to using veterinary medicines. Long-term observations are made of the animals to manage optimal birthing and punctual treatment is given to those needing more care than others due to the practice of semi-wild grazing [42].

As regards the soil, the operators stated that they have excellent chemical and biological quality and a high level of organic substances in more than 80% of the land used. They also underlined that activities have been started to reduce erosion such as minimum tillage, and that the ratio between land in excellent condition and degraded land was a positive one in favor of the former. The other

sub-theme analyzed was biodiversity. Although no real programs for the protection of threatened animal and plant species have been launched, particular attention was noted towards local species (both in rotation and in mixtures) and/or native animal breeds.

To achieve a more detailed analysis, dichotomization procedures were carried out on the affiliation matrix. This procedure involves assigning a connection value of 1 if there is sharing of an event and of 0 if there is no sharing.

In this way, an estimation of the dynamism of the network was achieved upon variation of the dichotomization "cut off" simply by measuring the density of the network itself (Figure 2). Density means the ratio between the number of connections existing as a proportion of the number of possible connections (the extreme case is where all farms share all information). The "cut off" also offers the opportunity to distinguish the shared level of rating. A cut off of 1 indicates that there is a connection between two farms if they share information regardless of the level of rating quality. A cut off of 2 indicates that the connection exists if information is shared with a rating score of 2, 3, 4 or 5 (limited, moderate, good, best, excluding unacceptable). A cut off of 3 indicates that the connection exists if information is shared with a rating score of 3, 4 or 5 (moderate, good, best, excluding limited and unacceptable), etc.

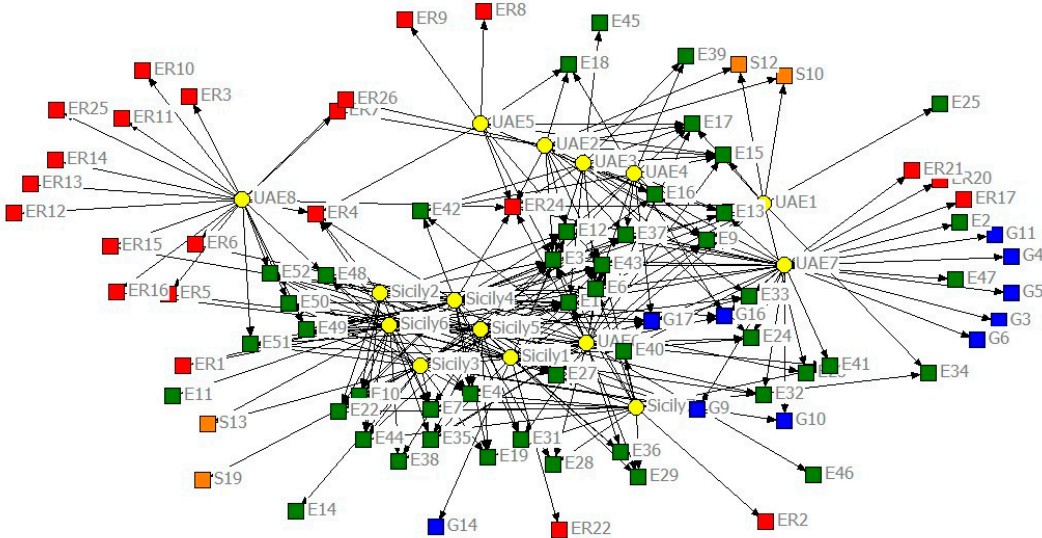

**Figure 2.** Dichotomized matrix with the critical issues of the network in the sample of organic farms that joined the study to measure sustainability with SAFA–FAO in the UAE and in Sicily. Italy (2017). Density means the ratio between the number of connections existing as a proportion of the number of possible connections (the extreme case is where all farms share all information). Yellow denotes the farms; red are the sustainability indicators for "Economic Resilience"; orange are the sustainability indicators for "Social Well-Being", green are the sustainability indicators for "Environmental Integrity"; blue are the sustainability indicators for "Good Governance".

It can be seen that networks connected through information shared with a "best" rating level represent only about 10% of the possible connections. This result necessitated taking the analyses to a deeper level by using different cut off values: (a) 1 if the rating score is 1 or 2; (b) 1 if the rating score is higher than 4; (c) 1 if the rating score is 5. In (a), organic agriculture we can observe the criticalities of the network, while in (b) and (c) we can observe the aspects of excellence of the network.

### 4.3. Analysis of Criticalities of the Network

The network exploits only about 17% of the possible connections, showing a low point in the average level of general criticality. The criticalities are related to the theme of "Environmental Integrity", and are strongly shared by farms in the Emirates. Thus, some farms predominantly show criticalities in

terms of "Economic Resilience", and in particular, for the aspects of "Investment" and "Vulnerability"; others show criticalities mainly in terms of "Economic Resilience", and in particular, for "Product Quality". "Information" and "Good Governance", with reference to the aspects of "Corporate Ethics" and "Accountability".

Considering only the shares. 62% of criticalities are related to the theme of "Environmental Integrity". 18% to the theme of "Social Well-Being". 14% to "Economic Resilience" and 6% to "Good Governance".

An "event by event" matrix was built, in which each element of the matrix is given by the sub-theme (indicated with its relevant ID) and the connection between the sub-themes is given by the number of shares. Then, an analysis of the network was carried out to identify the sub-themes in which most criticalities are concentrated and shared (Figure 3 and Table 4).

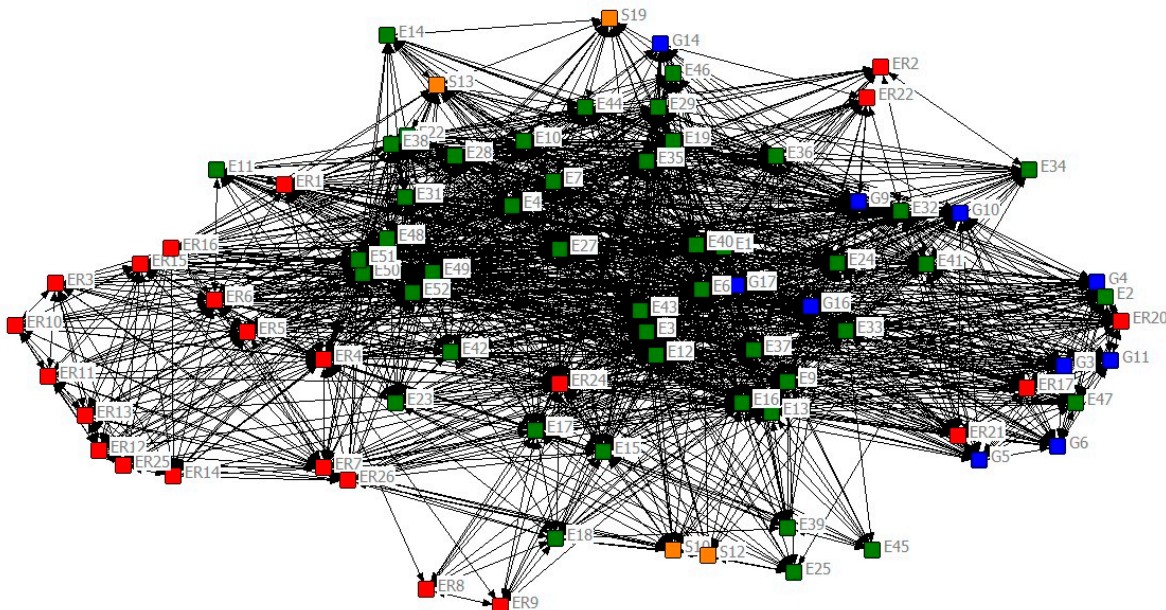

**Figure 3.** Matrix on the ties by critical sub-theme in the sample of organic farms that measured sustainability with SAFA–FAO in the UAE and in Sicily. Italy (2017). Each element of the matrix is given by the sub-theme (indicated with its relevant ID) and the connection between the sub-themes is given by the number of shares, an analysis of the network can be carried out to identify the sub-themes in which the most criticalities are concentrated and shared. Red denotes the sustainability indicators for "Economic Resilience"; orange are the sustainability indicators for "Social Well-Being", green are the sustainability indicators for "Environmental Integrity"; blue are the sustainability indicators for "Good Governance".

From this analysis we can see that the most shared criticalities generally relate to the context of "Environmental Integrity", and more specifically, that the most pressing issues relate to the "Atmosphere" (GHG Balance. Ambient Concentration of Air Pollutants. GHG Reduction Target and Air Pollution Reduction Target). "Animal Welfare" (Animal Health Practices. Animal Health. Humane Animal Handling Practices. Appropriate Animal Husbandry. Freedom from Stress) and "Water" (Water Conservation Target. Ground and Surface Water Withdrawals. Concentration of Water Pollutants).

To observe how shares between criticalities are distributed within each context identified by the SAFA questionnaire, frequency distributions were calculated (Figure 4). Each bin (with a width of 10) indicates the range of importance (degree values) and the frequency indicates the number of times that they fall within the specific "criticality".

**Table 4.** Measurement of the degree of centrality by sub-theme of criticality through the "Freeman's degree" index in the sample of organic farms that measured sustainability with SAFA–FAO in the UAE and in Sicily. Italy (2017).

| ID | Degree | ID | Degree | ID | Degree | ID | Degree | ID | Degree | ID | Degree | ID | Degree |
|------|--------|------|--------|------|--------|------|--------|------|--------|------|--------|------|--------|
| E3 | 249 | E52 | 132 | E19 | 87 | E36 | 65 | ER7 | 33 | ER20 | 28 | ER12 | 19 |
| E6 | 226 | E10 | 122 | G16 | 86 | E41 | 65 | ER26 | 33 | ER21 | 28 | ER13 | 19 |
| E43 | 194 | E9 | 118 | E33 | 86 | E28 | 64 | ER15 | 31 | E11 | 27 | ER14 | 19 |
| E1 | 183 | E7 | 116 | E16 | 82 | E29 | 62 | G3 | 28 | S10 | 26 | ER22 | 19 |
| E40 | 152 | E12 | 112 | E15 | 80 | ER1 | 50 | G4 | 28 | S12 | 26 | ER25 | 19 |
| E4 | 143 | G17 | 109 | E17 | 80 | G10 | 47 | G5 | 28 | E39 | 24 | S19 | 19 |
| ER24 | 142 | E22 | 103 | E24 | 80 | ER6 | 46 | G6 | 28 | E46 | 21 | ER2 | 16 |
| E27 | 134 | E37 | 102 | E13 | 79 | ER16 | 42 | G11 | 28 | G14 | 19 | E25 | 12 |
| E48 | 132 | E44 | 95 | E31 | 73 | S13 | 42 | E2 | 28 | E14 | 19 | E45 | 12 |
| E49 | 132 | G9 | 91 | E32 | 72 | ER5 | 40 | E34 | 28 | ER3 | 19 | ER8 | 9 |
| E50 | 132 | ER4 | 90 | E42 | 72 | E23 | 39 | E47 | 28 | ER10 | 19 | ER9 | 9 |
| E51 | 132 | E35 | 88 | E38 | 69 | E18 | 35 | ER17 | 28 | ER11 | 19 | ER12 | 19 |

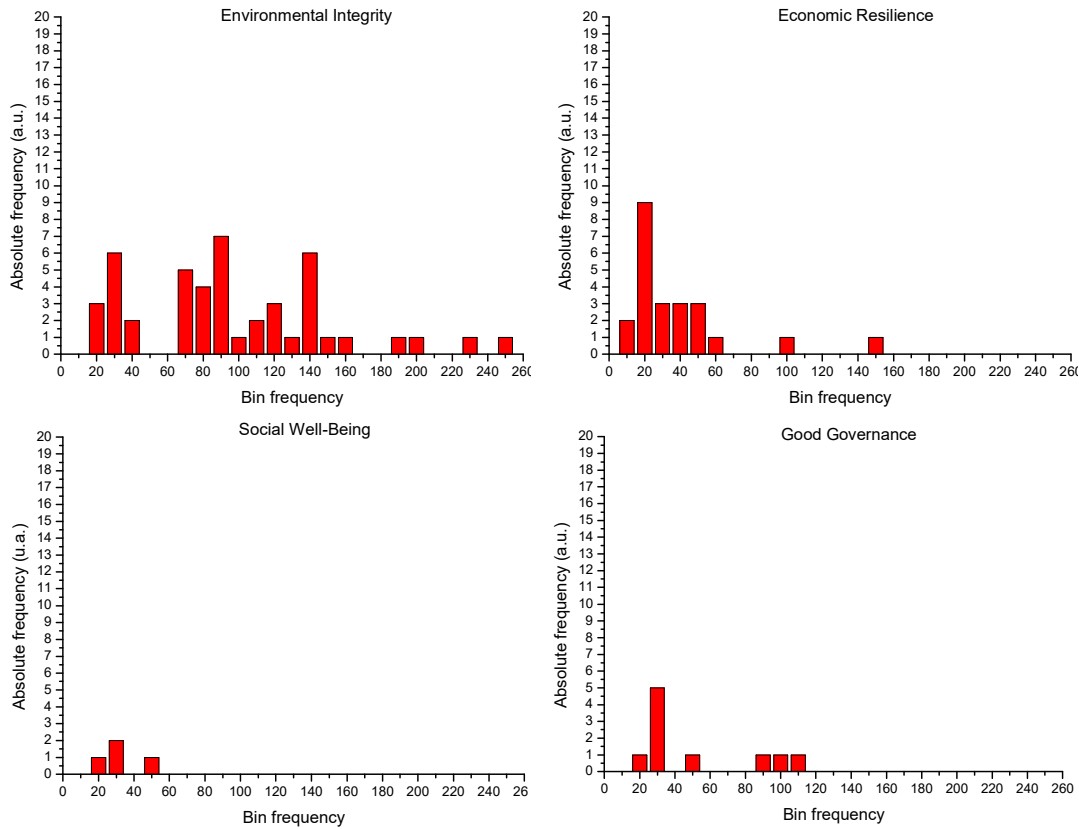

**Figure 4.** Critical and critical frequency ranges through the "Freeman's degree" index in the sample of organic farms that measured sustainability with SAFA–FAO in the UAE and in Sicily. Italy (2017). Each bin (with a width of 10) indicates the range of importance and the frequency indicates the number of times that they fall within the specific "criticality".

For each context or macro-area ("Good Governance" (also "corporate" ethics). "Environmental Integrity". "Economic Resilience" and "Social Well-Being"), the average weighted importance value of a specific context was calculated with the following formula:

$$\text{Weight Importance Value (WIV)} = \frac{\sum_i \text{BCV}_i \times \text{AFV}_i}{\sum_i \text{AFV}_i} \tag{5}$$

where BCV is the bin central value and AFV is the absolute frequency value.

The results obtained are reported in Table 5.

**Table 5.** Weight importance value (WIV) for each context or macro-area.

| Contextualization | WIV | WIV % |
| --- | --- | --- |
| Economic Resilience | 92 | 46.1 |
| Environmental Integrity | 33 | 16.5 |
| Social Well-Being | 28 | 13.8 |
| Good Governance | 47 | 23.6 |

*4.4. Network Criticalities—Networks between Actors*

In order to identify which farm has a central role in the network created according to shared criticalities, an actor-actor matrix was constructed based on the dichotomized one, in which each element of the matrix represents the number of shares that each actor has with the others. Sharing, as shown in Figure 5, is represented by a connection, the length of which is inversely proportional to its intensity.

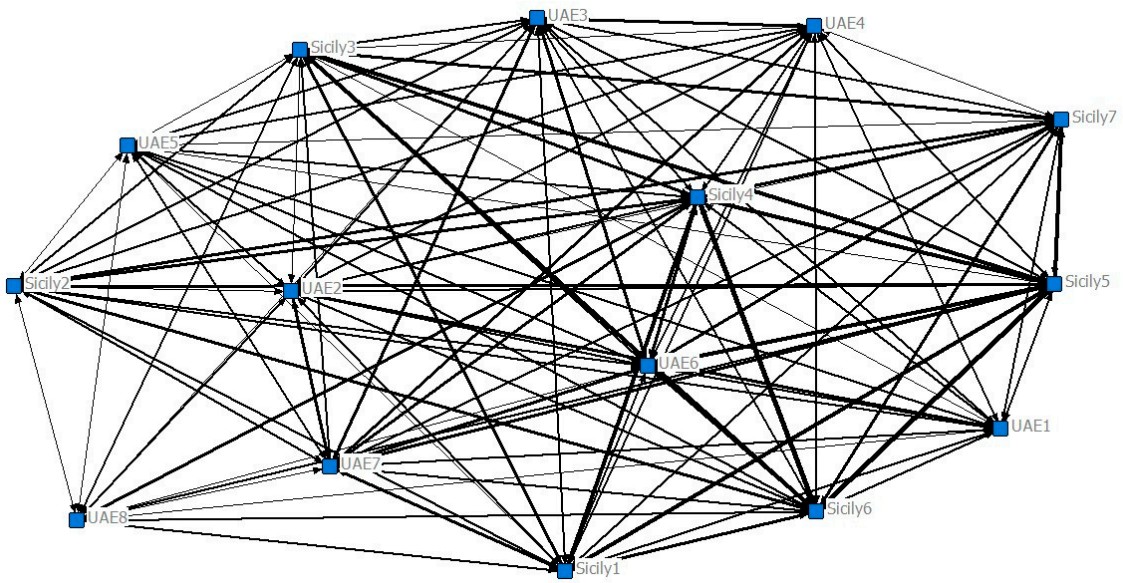

**Figure 5.** Actors-actors matrix for the identification of the central role in the network created on the shared criticalities in the sample of organic farms that measured sustainability with SAFA–FAO in the UAE and in Sicily. Italy (2017).

The density (98%) shows that nearly all the possible connections are used, i.e., all the actors share all the criticalities. The average geodetic distance is 1.019 and the diameter is 2 (Table 6). The first value indicates that within the network, on average, two adjacent nodes communicate directly with each other and do not always have an intermediary whereas the value 2 indicates that there are nodes that communicate via an intermediary. The data relating to the geodetic distances, together with the low level of centralization (0.022) indicate that the criticalities are evenly distributed across the network.

**Table 6.** Results of the model of centrality in the actors-actors matrix.

| Variable | Value | Variable | Value |
| --- | --- | --- | --- |
| Density | 0.981 | Diameter | 2 |
| Avg Geodetic Distance | 1.019 | Degree Centralization | 0.022 |

By studying the degree of centrality of the individual actors, it can be deduced that the two farms with the lowest degree of centrality (0.157 and 0.184, respectively) are located in the UAE. As shown previously, these are also the two farms that show unique criticalities compared to the others.

### 4.5. Analysis of Aspects of Excellence of the Network

The elements of sustainability upon which high levels of sharing were obtained among the farms sampled are summarized in the dichotomized matrix type (b) and (c), shown in Figure 6. In this, and particularly in the type (b) network graph, it emerges that the farms are very close, showing a shared rating score of 4 and 5 except for on issues regarding "Environmental Integrity" and "Good Governance", which have a marginal position. These issues, as shown by the type (c) network graph raising the value of excellence, are closely connected only in one case in the UAE.

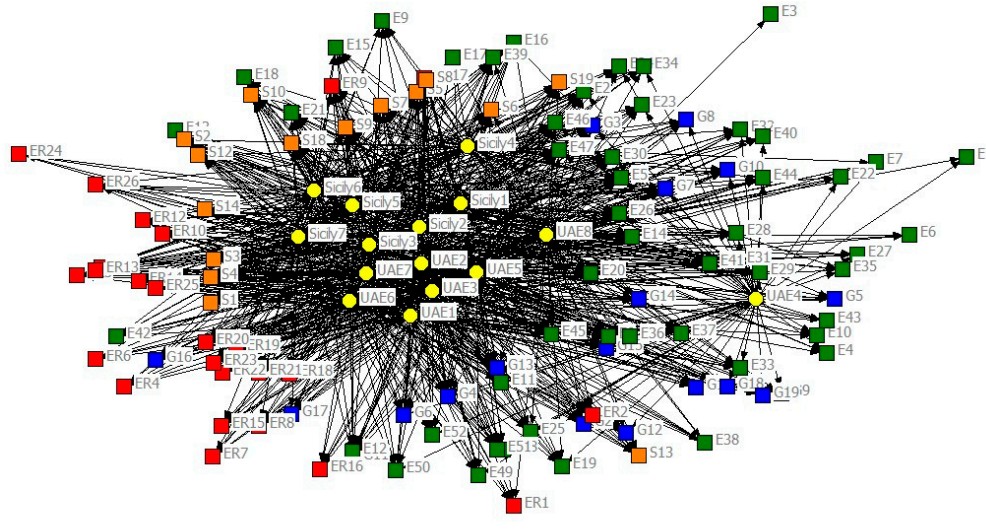

**Type network (b).** A binding value of 1 is assigned when the rating is 4 or 5. Value 0 when the rating values are different from 4. 0. 5.

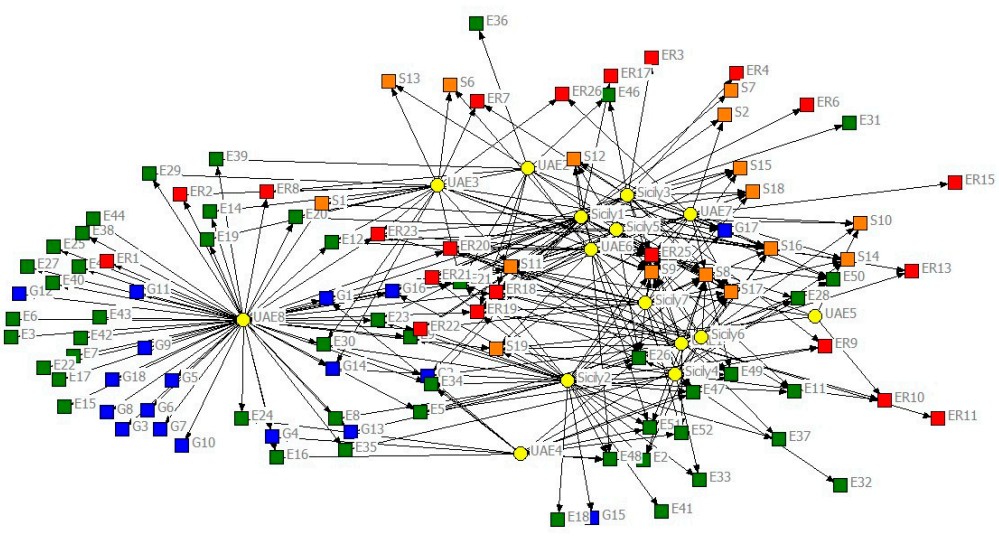

**Type network (c).** A binding value of 1 is assigned when the rating is 5. Value 0 when the rating values are different from 5.

| Typology | Density | SD |
|----------|---------|--------|
| Type B network | 0.6844 | 0.4648 |
| Type C network | 0.1811 | 0.3851 |

**Figure 6.** Graph of the dichotomized matrix of type (**b**) and (**c**) in the sample of organic farms that have measured sustainability with SAFA–FAO in the UAE and in Sicily. Italy (2017).

The density values relating to the two networks confirm a reduction in shares between actors and aspects of excellence as the rating score increases. In fact, this goes from 0.6844 in density (about 68% of possible connections) for ratings of 4 or 5, to 0.1811 in density (about 18% of possible connections) for a rating of 5. To identify on which sub-themes the aspects of excellence are concentrated and shared, an "event by event" matrix was created for case (c) (Figure 7), and the relevant degree of centrality was measured using the Freeman's degree index (Table 7).

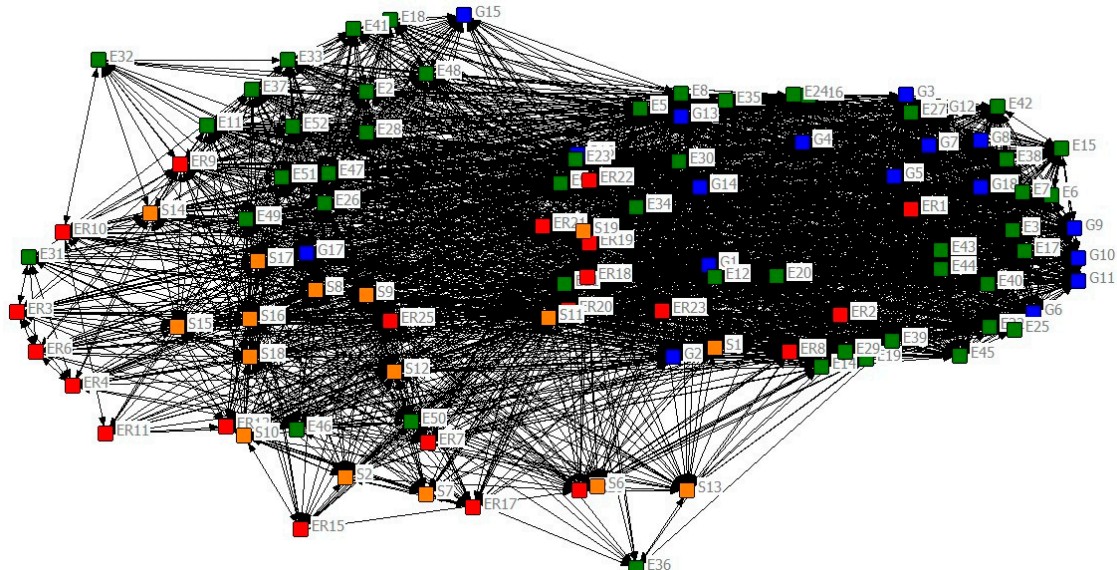

**Figure 7.** "Event x event" matrix aimed at identifying the sub-themes where the greatest excellence is concentrated and shared in the sample of organic farms that measured sustainability with SAFA–FAO in the UAE and in Sicily. Italy (2017).

**Table 7.** Freeman's degree index for the measurement of the degree of centrality for each sub-theme of excellence in the sample of organic farms that measured sustainability with SAFA–FAO in the UAE and in Sicily. Italy (2017).

| ID | Degree | ID | Degree | ID | Degree | ID | Degree | ID | Degree | ID | Degree | ID | Degree | ID | Degree |
|---|---|---|---|---|---|---|---|---|---|---|---|---|---|---|---|
| ER18 | 214 | E34 | 128 | E49 | 93 | E2 | 74 | G8 | 57 | G6 | 57 | ER13 | 40 | ER3 | 23 |
| ER25 | 206 | S16 | 121 | E20 | 92 | E35 | 73 | E25 | 57 | G10 | 57 | ER10 | 37 | E32 | 16 |
| S9 | 206 | E5 | 118 | E47 | 91 | ER9 | 73 | E40 | 57 | E6 | 57 | ER26 | 37 | ER15 | 16 |
| E21 | 193 | G16 | 114 | G13 | 91 | E16 | 70 | G3 | 57 | G11 | 57 | S6 | 35 | ER11 | 16 |
| S11 | 192 | E23 | 114 | E8 | 91 | E50 | 70 | G18 | 57 | E15 | 57 | G15 | 34 | E36 | 11 |
| S8 | 187 | E9 | 114 | ER8 | 89 | G4 | 70 | G5 | 57 | E38 | 57 | E18 | 34 | - | - |
| ER19 | 184 | S12 | 112 | E52 | 80 | S14 | 70 | E43 | 57 | G7 | 57 | E41 | 34 | - | - |
| ER20 | 162 | E30 | 112 | E19 | 78 | E24 | 70 | E17 | 57 | G9 | 57 | E46 | 34 | - | - |
| S17 | 160 | G14 | 112 | E29 | 78 | E39 | 68 | E7 | 57 | S10 | 56 | S13 | 32 | - | - |
| S19 | 158 | G2 | 110 | ER2 | 78 | E48 | 61 | E27 | 57 | E28 | 55 | ER17 | 24 | - | - |
| ER21 | 155 | E12 | 106 | E14 | 78 | E11 | 59 | E42 | 57 | E37 | 51 | S7 | 24 | - | - |
| ER23 | 133 | S1 | 102 | S15 | 78 | E3 | 57 | ER1 | 57 | E33 | 50 | E31 | 23 | - | - |
| ER22 | 131 | G1 | 94 | G17 | 78 | E22 | 57 | G12 | 57 | ER7 | 44 | ER6 | 23 | - | - |
| E26 | 129 | S18 | 94 | E51 | 77 | E44 | 57 | | | S2 | 40 | ER4 | 23 | - | - |

From the table it can be deduced that the most shared aspects of excellence are generally related to the context of "Economic Resilience", and specifically, the most prominent issues are Product Quality and Information (Hazardous Pesticides. Food Contamination. Food Contamination. Product Labeling. Traceability System. Certified Production); "Social Well-Being", specifically, the main themes of Labor Rights (Forced Labor. Child Labor). Equity (Non-Discrimination). Human Safety and Health (Public Health); Environmental Integrity and specifically, the topics related to Land (Net Gain/Loss of Productive Land. Land Use and Land Cover Change).

As for "economic resilience", the results highlighted excellent results among arable farms (Figure 8).

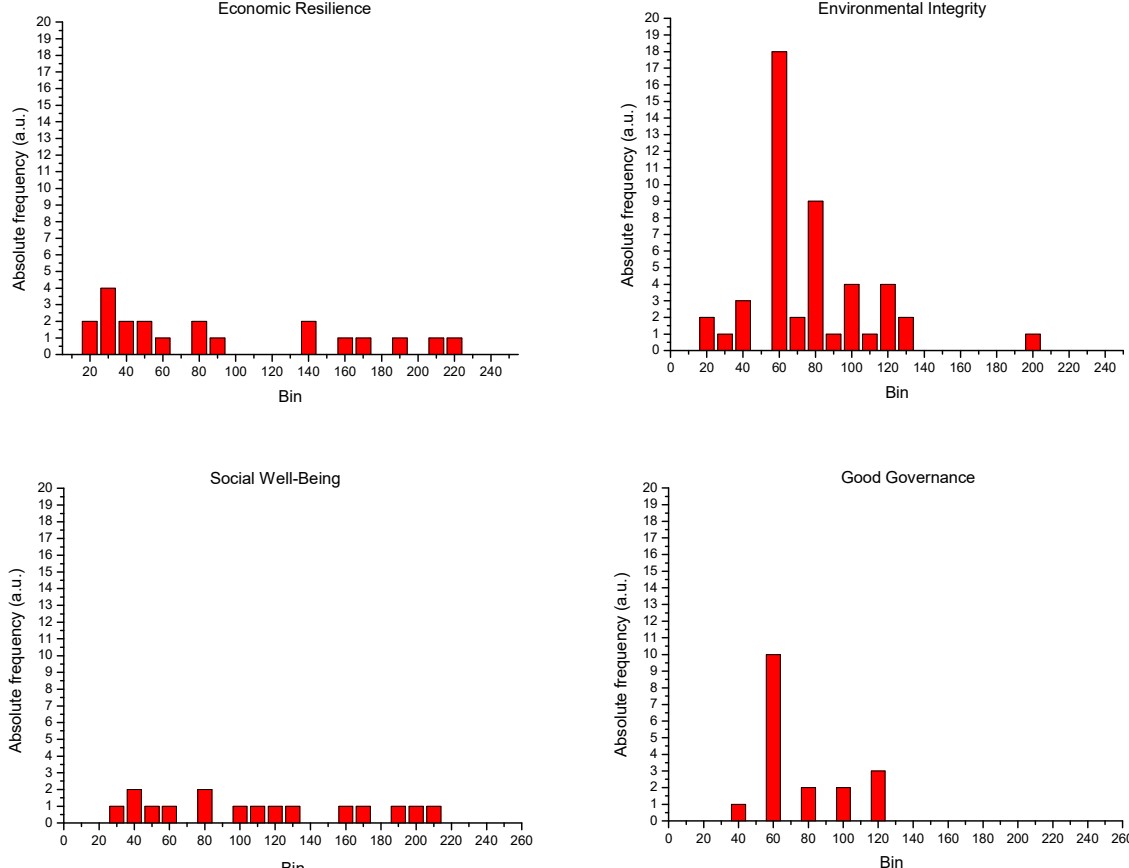

**Figure 8.** Frequency distributions by macro themes of excellence in the sample of organic farms that measured sustainability with SAFA–FAO in the UAE and in Sicily. Italy (2017).

For each context, the weight importance value (WIV) was calculated (see Section 4.3). The results obtained are reported in Table 8.

**Table 8.** Weight importance value (WIV) for each context or macro-area.

| Contextualization | WIV | WIV % |
|---|---|---|
| Economic Resilience | 45 | 22.8 |
| Environmental Integrity | 38 | 19.5 |
| Social Well-Being | 55 | 28.2 |
| Good Governance | 58 | 29.5 |

In both areas, operators said that they had invested to improve the production process, for example, by renewing their fleet of machinery and/or acquiring new land in order to increase their production capacity. In these cases, farms had access to formal sources of finance (often the EU Rural Program. Sicily 2007–2013 and resources specifically for the UAE) as well as informal sources (self-financing). They appeared able to sustain conditions of risk, and therefore, were suitable for the area of vulnerability. Excellent results were also obtained as regards local development, given that farm products are often destined to regional markets and are also characterized by a meticulous control process and a precise traceability system. As regards livestock farms, in this case too, the economic area showed medium-high results. All the operators stated that they had made investments, mainly in milking equipment, but also in improving the shelters used during the colder periods of the year. These farms also reported having access to sources of finance for tackling difficult situations, and in terms of product quality they all submit themselves to a meticulous system of control, certification and traceability. All the livestock farms evaluated in the study sell their products in local markets, and

unlike arable farms that are largely family-run, they take on seasonal workers, thus, contributing to the creation of both local and foreign employment.

Finally, in terms of "social well-being" the results reported by farms for this area of sustainability were medium-high, since attention is paid to ensuring adequate sustenance for both workers and for the enterprise owner. Better results were also measured for the theme of "fair trade practices", where enterprise owners stated that they maintain relationships with 100% of their suppliers and customers based on fair contracts that make it possible to sell the product easily to market, though at an unsatisfactory price for wheat.

The operators also demonstrated appropriate respect for workers' rights. These are enterprises that support female employment in this sector, and in the future, they would like to make a commitment to creating suitable facilities to accommodate workers with disabilities, thus, providing adequate jobs and including this group in the sector.

### 4.6. Network Aspects of Excellence—Networks between Actors

In order to identify which farm has a central role in network type (c) created according to shared aspects of excellence, an actor-actor matrix was constructed based on the dichotomized one, in which each element of the matrix represents the number of shares that each actor has with the others. Sharing, as shown in the following graph, is represented by a connection the length of which is inversely proportional to its intensity. Its thickness also indicates its intensity (Figure 9).

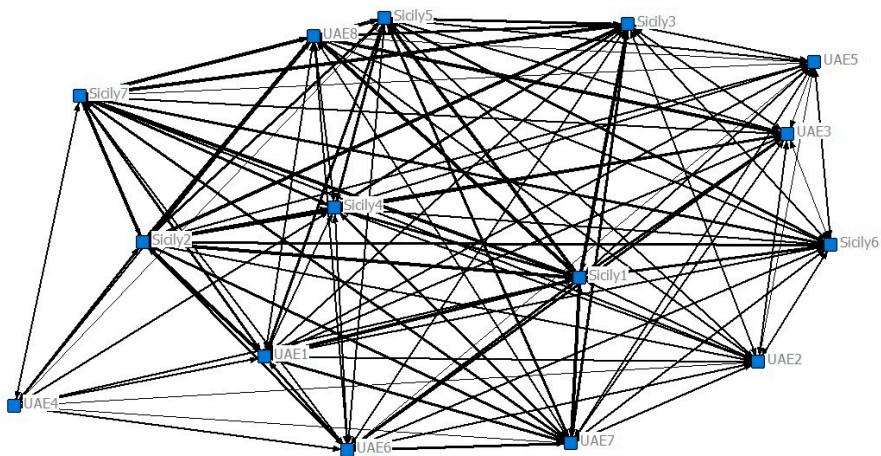

**Figure 9.** Matrix actors-actors and shares for excellence factors in the sample of organic farms that measured sustainability with SAFA–FAO in the UAE and in Sicily. Italy (2017).

The density (97%) shows that nearly all the possible connections are used, i.e., all the actors share a certain number of aspects of excellence in the same way. The average geodetic distance is 1.029 and the diameter is 2. The data relating to the geodetic distances, together with the low level of centralization (0.033) indicates that the number of aspects of excellence are evenly distributed between the actors.

Studying the degree of centrality of the individual actors in particular, it emerged that farms. Sicily 4 and Sicily 7 have the highest degree of centrality, while farms. UAE 6 and UAE 7 have a marginal role in terms of centrality, as shown in Table 4.

## 5. Potentiality and Limits of the Methodological Tools

SAFA has proved to be a powerful tool for measuring sustainability in agri-food chains thanks to the variety of indicators it provides. At the same time, it is very flexible because it makes it possible to select indexes according to the characteristics of the company and the aims of the analysis (one or more phases of the supply chain). These aspects are important, considering the open source nature of this resource. However, some aspects need to be improved; on the one hand, its functionality is reduced

because it requires an average interview time of at least 6–8 h, and on the other, there is wide qualitative variability in the accepted answers. For some indicators this is strictly linked to the availability of accounting and/or non-accounting data from the countryside notebook, the phytosanitary booklet, etc, and as such, data collection is required in preparation for the interview. For other indicators, however, the qualitative response and attribution of the level of accuracy of the score is at the discretion of the interviewer. Therefore, basic preparation of the interviewers who will carry out the survey is required to at least maintain the same yardstick.

The SNA, on the other hand, is recognized as a well-tested tool for analyzing relationships in the literature. It is very effective for defining and understanding formal and informal interactions between different roles and subjects within the same organizational context.

In the context of this research, the analysis of social networks made it possible to study the collaborative structure between companies and the relationships they have (directly and/or indirectly) for the exchange of knowledge and adherence to the measurement system of SAFA sustainability, as an innovative model to achieve positive impacts on the area and on the production units themselves.

The results showed that the network, even informally, through relations of influence, allows an increase in moments of confrontation and mediation between farmers and researchers.

However, there are some misalignments between collaboration, exchange of knowledge and networks of influence (demonstrated by weak links with more influential companies). A possible explanation of the poor connection, and therefore, of poor sharing, may be due to the presence in the sample of companies with profoundly different technical-production problems (on the one hand animal husbandry, and on the other, fruit and vegetables, for example), which could negatively affect the identification of possible collaborative "best practices" towards achieving a higher level of sustainability (in fact, the convergence towards the various indicators of the SAFA demonstrate this). The reality is different within the same production sector, where it is possible to identify companies with a central position in the network, being sought for collaboration in the adoption of innovative processes.

The effects of the influence of the network also differ from country to country (UAE and Sicily), from the role played by knowledge holders, by the willingness to share and the links with institutions and the research world. The latter, in fact, have an important role in their capacity to support innovation and willingness to collaborate, and in the governance on local basis, which is aimed at supporting farmers and other interested parties. It follows that some impacts on sustainability are not always due solely to the level of knowledge exchange (for example, the local adaptation of cultural practices, the management of working relationships, the job inclusion of subjects with low contracts, the solution of problems of pollution, etc.), but require institutional interventions for the removal of possible obstacles.

Ultimately, the results show the potential of social network analysis in identifying the strengths and limits of organic companies in terms of achieving impacts on holistic sustainability. The tendency towards collaboration is ethically inherent in the organic cultivation process, as is the attention to aspects and problems of the socio-economic, environmental and cultural life of the business, but the importance and the role of the actors should not be ignored. Institutions (local communities, research, subjects of the certification world, etc.) in collaborative networks, create space, experiment and legitimize new innovations towards sustainability.

## 6. Conclusions

Organic agriculture, with its close attention to the production system (resources, land, environment, etc.) and the socio-economic system, is a successful model of sustainable production in both the rich and poor countries of the planet.

Farmers make a commitment of ethical value and, through a process of conversion, adapt their business decision-making processes to a specifically prepared discipline and an institutional control system. By virtue of this choice they take on an active role within the areas that they belong to, reinforcing their relationships in the local socio-economic, environmental and cultural fabric. This system of relationships sometimes follows a direct and codified pathway (for example through

a network-based contract), and at others, develops in an indirect and random way, creating an "atmosphere" effect that develops and gives substance to the area.

These informal relationships within the network also end up having positive effects on overall sustainability, like those observed in the network of farms linked by the FAO's SAFA sustainability measurement system.

The wide range of sustainability indicators measured in organic agriculture through the SAFA also show both the criticalities and aspects of excellence that are partly influenced by the network effect.

These are often medium-sized farms (vegetable, arable and/or livestock) that despite having clearly defined their mission by converting to an organic farming system, lack the suitable tools for promoting this to all parties involved in the supply chain and the production process as a whole [43].

The same results emerged from the answers given by operators regarding the responsibility process, in that the enterprise owners did not always state that they were able, through their decision-making processes, to influence all parties involved in the production process, therefore, they appear unconvincing and unable to define their objectives clearly or with the use of suitable tools.

Another negative aspect of the evaluation involved the sphere of participation, where the operators demonstrated a clear lack of understanding of how to get the interested parties involved; this is due to an inability to start a dialog and to reach solutions in the event of conflicts [42].

Within the network, there has been an increasing awareness of the political management of sustainability, with more openness towards a systemic, complementary and unitary perception of the goals and activities to be pursued. Fundamental among these have been the transfer of scientific knowledge and of best agricultural practices, and the maintenance and development of "local systems" of production-distribution-consumption to maintain quality production while paying attention to sustainability. The role played by the market also remains crucial. The UAE has come to organic agriculture more recently, and by virtue of the high capitalization of farms and a scarcity in the availability of available natural resources, has shown a good overall ability to tackle the particular organizational and management problems specific to sustainable agriculture. Additionally, because of a high level of internal demand for organic products and strong interest from large-scale retailers, the drive towards organic production methods seems destined to increase in the future. This process, though with different mechanisms, also applies to Sicily, a land where conversion to organic methods is driven partly by the market and partly by the economic sustainability that these methods ensure in certain local contexts.

In conclusion, the work carried out shows that farms in the network can sustain themselves in the process of improving their sustainability performance in the case of organic production methods. This is because of the fact that, in addition to the environmental aspects, the network can work together in supporting the economic, social and cultural aspects of this process (e.g., materials and energy, investment, vulnerability, labor rights, equity, etc.).

In the future, organic farmers will need to focus more on consumption at the downstream stage in order to utilize the aspects of sustainability that are most perceived by consumers, and to benefit from their readiness to associate this value with the enterprise.

**Author Contributions:** The study is the result of full collaboration and therefore all authors accept full responsibility for it. The actual writing of sections "Introduction" is attributable to K.B.A.S.; the sections "Conceptual framework" and "References" are attributable to P.G.; the sections "Research design", "Analysis methodology", "Characteristics of the farms", "Analysis of criticalities of the network", "Analysis of aspects of excellence of the network" and "Potentiality and limits of the methodological tools" are attributable to G.T.; the sections "Measurement tools: the SAFA approach" and "Appendix A" are attributable to P.C.; the sections "Analysis of the affiliation network", "Network criticalities—network between actors" and "Network aspects of excellence—network between actors" are attributable to G.S.; the section "Conclusions" is attributable to S.L.C., and C.L.

**Funding:** This study was carried out thanks to the financial support of the Memorandum of Cultural and Scientific Cooperation between the "ABU DHABI DEVELOPMENT GROUP" (United Arab Emirates) and the University of Catania (Luciano Cosentino, scientific coordinator).

**Conflicts of Interest:** The authors declare no conflict of interest.

## Appendix A

**Table A1.** Sustainability indicators and IDs attributed in the SNA model used.

| Contextualization | Themes | Sub-themes | ID |
|---|---|---|---|
| GOOD GOVERNANCE | Corporate ethics | Mission Explicitness | G1 |
| | | Mission Driven | G2 |
| | | Due Diligence | G3 |
| | Accountability | Holistic Audits | G4 |
| | | Responsibility | G5 |
| | | Transparency | G6 |
| | Participation | Stakeholder Identification | G7 |
| | | Stakeholder Engagement | G8 |
| | | Engagement Barriers | G9 |
| | | Effective Participation | G10 |
| | | Grievance Procedures | G11 |
| | | Conflict Resolution | G12 |
| | Rule of Law | Legitimacy | G13 |
| | | Remedy, Restoration and Prevention | G14 |
| | | Civic Responsibility | G15 |
| | | Free, Prior and Informed Consent | G16 |
| | | Tenure Rights | G17 |
| | Holistic management | Sustainability Management Plan | G18 |
| | | Full-Cost Accounting | G19 |
| ENVIROMENTAL INTEGRITY | Atmosphere | GHG Reduction Target | E1 |
| | | GHG Reduction Target | E2 |
| | | GHG Balance | E3 |
| | | Air Pollution Reduction Target | E4 |
| | | Air Pollution Prevention Practices | E5 |
| | | Ambient Concentration of Air Pollutants | E6 |
| | Water | Water Conservation Target | E7 |
| | | Water Conservation Practices | E8 |
| | | Ground and Surface Water Withdrawals | E9 |
| | | Clean Water Target | E10 |
| | | Water Pollution Prevention Practices | E11 |
| | | Concentration of Water Pollutants | E12 |
| | | Wastewater Quality | E13 |
| | Land | Soil Improvement Practices | E14 |
| | | Soil Physical Structure | E15 |
| | | Soil Chemical Quality | E16 |
| | | Soil Biological Quality | E17 |
| | | Soil Organic Matter | E18 |
| | | Land Conservation and Rehabilitation Plan | E19 |
| | | Land Conservation and Rehabilitation Practices | E20 |
| | | Net Gain/Loss of Productive Land | E21 |

**Table A1.** *Cont.*

| Contextualization | Themes | Sub-themes | ID |
|---|---|---|---|
| ENVIROMENTAL INTEGRITY | Biodiversity | Landscape/Marine Habitat Conservation Plan | E22 |
| | | Ecosystem Enhancing Practices | E23 |
| | | Structural Diversity of Ecosystems | E24 |
| | | Ecosystem Connectivity | E25 |
| | | Land Use and Land Cover Change | E26 |
| | | Species Conservation Target | E27 |
| | | Species Conservation Practices | E28 |
| | | Diversity and Abundance of Key Species | E29 |
| | | Diversity of Production | E30 |
| | | Wild Genetic Diversity Enhancing Practices | E31 |
| | | Agro-biodiversity in-situ Conservation | E32 |
| | | Locally Adapted Varieties/Breeds | E33 |
| | | Genetic Diversity in Wild Species | E34 |
| | | Saving of Seeds and Breeds | E35 |
| | Materials and Energy | Material Consumption Practices | E36 |
| | | Nutrient Balances | E37 |
| | | Renewable and Recycled Materials | E38 |
| | | Intensity of Material Use | E39 |
| | | Renewable Energy Use Target | E40 |
| | | Energy Saving Practices | E41 |
| | | Energy Consumption | E42 |
| | | Renewable Energy | E43 |
| | | Waste Reduction Target | E44 |
| | | Waste Reduction Practices | E45 |
| | | Waste Disposal | E46 |
| | | Food Loss and Waste Reduction | E47 |
| | Animal Welfare | Animal Health Practices | E48 |
| | | Animal Health | E49 |
| | | Humane Animal Handling Practices | E50 |
| | | Appropriate Animal Husbandry | E51 |
| | | Freedom from Stress | E52 |
| ECONOMIC RESILIENCE | Investment | Internal investment | ER1 |
| | | Community Investment | ER2 |
| | | Long Term Profitability | ER3 |
| | | Business Plan | ER4 |
| | | Net Income | ER5 |
| | | Cost of Production | ER6 |
| | | Price Determination | ER7 |
| | Vulnerability | Guarantee of Production Levels | ER8 |
| | | Product Diversification | ER9 |
| | | Procurement Channels | ER10 |
| | | Stability of Supplier Relationships | ER11 |

**Table A1.** *Cont.*

| Contextualization | Themes | Sub-themes | ID |
|---|---|---|---|
| ECONOMIC RESILIENCE | Vulnerability | Dependence on the Leading Supplier | ER12 |
| | | Stability of Market | ER13 |
| | | Net Cash Flow | ER14 |
| | | Safety Nets | ER15 |
| | | Risk Management | ER16 |
| | Product Quality and Information | Control Measures | ER17 |
| | | Hazardous Pesticides | ER18 |
| | | Food Contamination | ER19 |
| | | Food Quality | ER20 |
| | | Product Labeling | ER21 |
| | | Traceability System | ER22 |
| | | Certified Production | ER23 |
| | Local Economy | Regional Workforce | ER24 |
| | | Fiscal Commitment | ER25 |
| | | Local Procurement | ER26 |
| SOCIAL WELL-BEING | Decent Livelihood | Right to quality of life | S1 |
| | | Wage Level | S2 |
| | | Capacity Development | S3 |
| | | Fair Access to Means of Production | S4 |
| | Fair Trading Practices | Fair pricing and transparent contracts | S5 |
| | | Rights of Suppliers | S6 |
| | Labor Rights | Employment Relations | S7 |
| | | Forced Labor | S8 |
| | | Child Labor | S9 |
| | | Freedom of Association and Right to Bargaining | S10 |
| | Equity | Non-Discrimination | S11 |
| | | Gender Equality | S12 |
| | | Support to Vulnerable People | S13 |
| | Human Safety and Health | Safety and Health Trainings | S14 |
| | | Safety of Workplace, Operations and Facilities | S15 |
| | | Health Coverage and Access to Medical Care | S16 |
| | | Public Health | S17 |
| | Cultural Diversity | Indigenous Knowledge | S18 |
| | | Food Sovereignty | S19 |

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
