# Peer review of "Analysis of Relationships and Sustainability Performance in Organic Agriculture in the United Arab Emirates and Sicily (Italy)"

_resources, doi:10.3390/resources8010039_

Reviewer 1 Report

The thematic is very interesting and the paper is well structured and innovative. However some comments and suggestions should be considered for the improvement of the paper.

(1) In the “Conceptual framework” is presented some research on the theme, but the gathered results or learnings in that studies are very little explored (e.g. from line 130 to 150).

For this point I suggest other readings as the paper: Sala, S.; Ciuffo, B; Nijkamp, P. (2015). A Systematic framework for sustainaibility assessment. Ecological Economics 119 (314-325). 

(2) In the "Analysis methodology" all the variables of the equations inserted in the text should appear in italic. Only the variables exposed in lines 287 and 288 use that form.

 (3) Some times the authors use United Arab Emirates others UAE.

 (4) Table 2 should be restructured. Last column is not perceptible and it is very difficult to follow the information in all the rows. For example what are the correspondent farms to the “X” of the “diversification of productive activities”?

"X” in type of production should be replaced by “% of surface” for each farm. Regarding the livestock, more information (as species and number of heads) should be exposed.

Yet in this table, in the column “diversification of productive activities” some “X” need explanation. For example, farm 2 has “X” but it has only cereals and livestock, however farm 3 has not “X”, but it has 4 types of production.

 (5) More  information has to be given for the interpretation and the following of the "analysis of the affiliation network". Also there are missing the presentation of the results to support the figures and its interpretation and analysis.  

How were Figures 4 and 8 constructed?

(6) For a better comparison and interpretation of the results, I suppose that not organic farms should be used. 

 (7) The software used for the graphs should be presented.

Author Response

1 We have read and inserted the suggested paper

2 Revision made in italics

3 We used the acronym after the first citations

4 We have modified table 2 welcoming the suggestions. Now it is readable

5 We have included explanations for the processing done and two explanatory tables

6 we have modified the text

7 the reference to the software used has been inserted

Thanks for the valuable suggestions

Reviewer 2 Report

The authors present an interesting and relevant study, in the field.

 From my point of view, the study is novelty and contribute to the literature on organic agriculture.

I think the title is very clear and the abstract reflects adequately the content of the article.

The paper is well structured and documented and methodology is clearly presented.

The conclusions are supported by the results and are useful for both theory and practice. 

Author Response

Thank you for the positive evaluation and the valuable suggestions